# Defining Transcriptomic Heterogeneity between Left and Right Ventricle-Derived Cardiac Fibroblasts

**DOI:** 10.3390/cells13040327

**Published:** 2024-02-10

**Authors:** Michael Bradley Dewar, Fahad Ehsan, Aliya Izumi, Hangjun Zhang, Yu-Qing Zhou, Haisam Shah, Dylan Langburt, Hamsini Suresh, Tao Wang, Alison Hacker, Boris Hinz, Jesse Gillis, Mansoor Husain, Scott Patrick Heximer

**Affiliations:** 1Department of Physiology, University of Toronto, Toronto, ON M5G 1M1, Canada; 2Translational Biology and Engineering Program, Ted Rogers Centre for Heart Research, University of Toronto, Toronto, ON M5G 1M1, Canada; 3Institute of Biomaterial & Biomedical Engineering, University of Toronto, Toronto, ON M5G 1M1, Canada; 4Toronto General Hospital Research Institute, University Health Network, Toronto, ON M5G 2C4, Canada; 5Ted Rogers Centre for Heart Research, Toronto, ON M5G 1M1, Canada; 6Keenan Research Institute for Biomedical Science of the St. Michael’s Hospital, Toronto, ON M5B 1W8, Canada; 7Faculty of Dentistry, University of Toronto, Toronto, ON M5G 1M1, Canada

**Keywords:** cardiac fibroblast subpopulations, myofibroblast transdifferentiation, ventricular differences, RNA sequencing, pressure overload

## Abstract

Cardiac fibrosis is a key aspect of heart failure, leading to reduced ventricular compliance and impaired electrical conduction in the myocardium. Various pathophysiologic conditions can lead to fibrosis in the left ventricle (LV) and/or right ventricle (RV). Despite growing evidence to support the transcriptomic heterogeneity of cardiac fibroblasts (CFs) in healthy and diseased states, there have been no direct comparisons of CFs in the LV and RV. Given the distinct natures of the ventricles, we hypothesized that LV- and RV-derived CFs would display baseline transcriptomic differences that influence their proliferation and differentiation following injury. Bulk RNA sequencing of CFs isolated from healthy murine left and right ventricles indicated that LV-derived CFs may be further along the myofibroblast transdifferentiation trajectory than cells isolated from the RV. Single-cell RNA-sequencing analysis of the two populations confirmed that *Postn*+ CFs were more enriched in the LV, whereas *Igfbp3*+ CFs were enriched in the RV at baseline. Notably, following pressure overload injury, the LV developed a larger subpopulation of pro-fibrotic *Thbs4*+/*Cthrc1*+ injury-induced CFs, while the RV showed a unique expansion of two less-well-characterized CF subpopulations (*Igfbp3*+ and *Inmt*+). These findings demonstrate that LV- and RV-derived CFs display baseline subpopulation differences that may dictate their diverging responses to pressure overload injury. Further study of these subpopulations will elucidate their role in the development of fibrosis and inform on whether LV and RV fibrosis require distinct treatments.

## 1. Introduction

Cardiac fibrosis, characterized by chronic and excessive deposition of extracellular matrix (ECM) throughout the heart, is a major risk factor for the development of heart failure. This phenomenon occurs primarily through the actions of resident cardiac fibroblasts (CFs), which become activated following cardiac injury and upregulate the secretion of ECM and other pro-fibrotic proteins [1]. Different pathophysiologic stimuli can trigger CF activation in the left ventricle (LV) and/or right ventricle (RV), creating distinct patterns of fibrosis throughout the myocardium [2].

Given the central role of CFs in the development of fibrosis and the impact of fibrosis on disease prognoses, CF characterization has become a focal point of cardiovascular research. Specifically, recent work has highlighted the extensive heterogeneity within CFs that was previously unappreciated. CFs isolated from the atria versus ventricles, and the left versus right atria, differ in fibrosis-related pathways such as proliferation rate, collagen production, and how effectively collagen production can be inhibited [3,4]. Single-cell RNA sequencing (scRNA-seq) has been particularly influential, revealing numerous CF subpopulations within healthy and diseased states. Farbehi et al. performed unbiased clustering of cardiac interstitial cells and identified eight distinct subpopulations at days 3 and 7 post-myocardial infarction (MI) [5]. These findings were extended by Ruiz-Villalba et al., who identified one of ten CF clusters (*Cthrc1*+ CFs) as critical for maintaining myocardial integrity after MI [6]. Lastly, MacLennan et al. reported eight distinct subpopulations and highlighted transcriptomic differences in CFs exposed to Angiotensin II infusion versus MI-induced cardiac injury [7]. 

Despite the previous reports of transcriptomic heterogeneity within CFs, no studies have directly compared CF heterogeneity in the LV versus RV. The two ventricles are distinct in many aspects, offering numerous sources of potential CF heterogeneity. During cardiogenesis, the LV originates from the first heart field, whereas the RV is derived from the secondary heart field [8]. The LV also experiences a higher-pressure environment as the systemic pump compared with the RV, which encounters lower pressures as the pulmonary pump. Anatomical differences further distinguish the two, with variations in their shape, size, orientation and number of muscle fibers, and resistance to ischemia [9]. Based on the distinct nature of the ventricles, we hypothesized that LV- and RV-derived CFs display transcriptomic heterogeneity that influences their proliferation and differentiation following cardiac injury.

Herein, we present RNA sequencing data from murine LV- and RV-derived CFs, highlighting gene expression and subpopulation differences. The top differentially expressed genes between these CFs showed strong co-expression that was conserved across mammals, including humans. We further identified two subpopulations, marked by *Igfbp3* and *Postn*, which vary in abundance across the uninjured ventricles. Additional chamber-specific differences in *Thbs4*+ and *Inmt*+ CF subpopulations were revealed following pressure overload injury. Finally, we report on sex-specific differences between LV- and RV-derived CFs in both the uninjured heart and pressure overload conditions. Collectively, these findings will inform on how inherent differences in LV- and RV-derived CFs affect the fibrotic response mounted by each ventricle following cardiac injury.

## 2. Methods

### 2.1. Mice

All animal work was conducted under the guidelines established by the Canadian Council on Animal Care (CCAC). All Experimental protocols were approved by the Institutional Animal Care and Use Committee at the University of Toronto and conducted in accordance with Canadian animal protection laws. Animals were housed in a pathogen-free environment at the University of Toronto’s Division of Comparative Medicine Animal Facility and had ad libitum access to water and standard rodent chow. All experiments were performed with *Col1a1-GFP* reporter mice, which have previously been described in depth [10].

### 2.2. Transverse Aortic Constriction and Pulmonary Artery Banding

Mice placed under isoflurane (2–3%) anesthesia were intubated and ventilated with the Hallowell mouse mini ventilator. In a sterile surgical environment, chest and abdominal hair were shaved and the underlying skin was wiped with chlorhexidine alcohol. For transverse aortic constriction (TAC), the thorax was opened by parasternal thoracotomy. Pulmonary artery banding (PAB) was performed using intercostal access to the thorax. The transverse aorta or pulmonary artery was dissected and separated from the adjacent structures before ligation with 6-0 silk. Ligation was performed by placing a 25–27-gauge needle on the artery for optimum constriction. The needle was quickly removed to keep the constricted artery patent. Before closing the chest, positive air pressure was supplied to the chest cavity to ensure the lungs remained inflated. The wound was then closed in layers by using 5-0 Prolene sutures. Animals undergoing sham surgery experienced the same procedure without artery constriction. To validate post-injury fibrosis, hearts were fixed with 4% paraformaldehyde and 5 um sections were obtained and stained with picrosirius red. Slides were imaged on an AxioImager 2 (Zeiss, Oberkochen, Baden-Württemberg, Germany). Body temperature was maintained using a heating pad.

### 2.3. Echocardiography

For all cardiac imaging of the studied mice, a high-frequency ultrasound imaging system (Vevo3100, FUJIFILM VisualSonics, Toronto, ON, Canada) with a 30 MHz transducer was used. Mice were placed under isoflurane anesthesia (induced at 3%, then maintained at 1.5%), then positioned supine with four paws taped to electrodes on a prewarmed platform for electrocardiogram (ECG) recording and heart rate monitoring. Mouse body temperature was monitored by a rectal thermometer and maintained around 37 °C. Chest and abdominal hair were removed using Nair hair-removal cream.

For analysis of TAC injury, the longitudinal section of the aortic arch was visualized from a right parasternal view as described previously [11]. The location of constriction was identified in 2D imaging and the stenotic flow jet was visualized by Doppler color flow mapping. Pulsed Doppler was used to record the flow velocity spectrum of the stenotic jet with a large sample volume located slightly distal to the site of constriction. The peak velocity and maximum pressure gradient were measured. In mice with sham surgery, Doppler color flow mapping indicated the location of the highest velocity in the lumen of the aortic arch. The pulsed Doppler sample volume was located to detect the highest velocity, usually close to the outer curvature at the level between the innominate artery and the left common carotid artery. For the left ventricular systolic function, the velocity and diameter of the aortic annulus were measured for calculating LV stroke volume and cardiac output. The left ventricle was also visualized along its long axis view and M-mode recordings made from its middle segment for measuring the anterior and posterior wall thicknesses and chamber dimensions. The fractional shortening and ejection fractions were calculated as described previously [12].

For PAB injury analysis, we followed a comprehensive imaging protocol that was recently established [13]. At first, the longitudinal sections from the right ventricular outflow tract to the main pulmonary artery were achieved through the left parasternal window for evaluating vessel dimensions and flow dynamics. In mice with PAB, B-mode imaging and Doppler color flow mapping were used to locate the site of constriction and the resultant stenotic flow jet. The pulsed Doppler sample volume was placed slightly distal to the site of constriction to record the stenotic flow jet during systole. The peak velocity (V_max_) of the stenotic jet was measured to calculate the maximal pressure gradient using a simplified Bernoulli equation: maximal pressure gradient (mmHg) = 4(V_max_)^2^. Then, the diameter of the pulmonary orifice at peak systole and the Doppler blood flow velocity spectrum at the same level were measured for calculating the right ventricular stroke volume and cardiac output. For evaluating the right ventricular remodeling, the right ventricular free wall thickness, inflow tract chamber dimension, and their dynamic change throughout the cardiac cycle were measured using anatomical M-mode recording from a left parasternal imaging window as previously described in detail [14].

### 2.4. Fibroblast Isolations

Following heart excision, the atria and atrioventricular valves were surgically removed. We then dissected the left and right ventricle free walls and washed the tissue in chilled DPBS. The remaining septal tissue was discarded. LV and RV tissue samples were then minced and digested in a solution of 2 mg/mL Collagenase type II (Cat#: LS004176, Worthington, Columbus, OH, USA) and 2 mg/mL Dispase II (Cat#: D4693, Sigma, St. Louis, MO, USA) at 37 °C for 45 min. Trituration was performed with a 10 mL serological pipette at the 15 and 30 min marks and with a 1 mL pipette at the 45 min mark. Samples were then filtered through 70 µm strainers into 30 mL of chilled DPBS, split into two 15 mL Eppendorf tubes, and centrifuged for 7 min at 500× *g*. After decanting, pellets were resuspended in 100 µL of Hank’s Balanced Salt Solution (HBSS) containing 2% FBS and incubated with CD31 and CD45 magnetic beads for 15 min. Endothelial (CD31+) and hematopoietic cells (CD45+) were subsequently removed using the AutoMacs instrument (Miltenyi Biotec, Bergisch Gladbach, North Rhine-Westphalia, Germany). The remaining cells were stained with DAPI, and live CFs (GFP^+^, DAPI^Neg^) were sorted on the Aria IIu (BD Biosciences, Franklin, NJ, USA) into HBSS containing 2% FBS for subsequent RNA isolation or preparation for scRNA-seq.

### 2.5. Bulk RNA Sequencing

RNA was isolated using the commercially available RNeasy mini kit (Cat#: 74004 Qiagen, Hilden, North Rhine-Westphalia, Germany) and RNase-Free DNase Set (Cat#: 79254, Qiagen, Hilden, North Rhine-Westphalia, Germany). Bulk RNA sequencing was performed at the Princess Margaret Genomics Centre using the Illumina NovaSeq6000 instrument, generating approximately 40 million paired-end reads per sample. Adaptor sequences were removed using cutadapt [15]. Reads were aligned using the pseudo alignment method from Kallisto [16]. The transcriptome used for pseudo alignment was obtained from the mouse genome build GRCm38 from ENSEMBL. Subsequent analyses were performed using R Statistical Software (v3.6.1 or v4.3.2) [17]. Transcript abundances were imported into R from Kallisto output files using Tximport [18]. The Limma package was used to quantify RNA counts and calculate technical and biological variability [19]. These were then used to generate test statistics based on a linear model, where the injury group was corrected against the intercept. Batch effects were present from the date of sample preparation and gene body coverage, both of which were incorporated into the differential expression model. Batch effects were removed from the normalized transcripts per million (TPM) data with the removeBatchEffects function in Limma before being used as input for batch-corrected heatmaps and PCA plots. Volcano plots were created using EnhancedVolcano [20].

### 2.6. Gene Set Enrichment Analysis

Full gene lists from differential gene expression testing were pre-ranked by beta value and used as input for gene set enrichment analysis (GSEA) through clusterProfiler [21], analyzing against Gene Ontology (GO) terms [22,23]. GSEA was run with a gene set size range minimum of 10 and maximum of 800, a *p*-value cutoff of 0.05, and a *p*-value adjustment using the Benjamini–Hochberg method. Results were then filtered by adjusted *p*-value (Padj < 10^−6^) to identify gene sets fit for further study. Dotplots of enriched gene sets were created using clusterProfiler.

### 2.7. Single-Cell RNA Sequencing

FACS-isolated CFs were manually counted using a hemocytometer before single-cell library preparation using the 10x Genomics Chromium system. Library preparation and sequencing were performed at The Centre for Applied Genomics at the Peter Gilgan Centre for Research and Learning. Approximately 10,000 cells were loaded onto the 10× Genomics Chromium Controller instrument to generate single-cell gel beads in emulsion (GEMs) using the Chromium Single Cell 3′ Reagent Kits v3 kit (part -1000092, 10× Genomics, Pleasanton, CA, USA) following the manufacturer’s instructions. Briefly, barcoded full-length cDNA is first produced from poly-adenylated mRNA. cDNA is then fragmented and made into Illumina-ready libraries. The final libraries contain a 10× barcode, UMI, sample index, and Illumina P5 and P7 sequences. The size, quality, and concentration of the cDNA and corresponding 10× library were evaluated by the Agilent 2100 Bioanalyzer system and quantified using the Kapa Library Quantification Illumina/ABI Prism Kit protocol (KAPA Biosystems, Wilmington, MA, USA). Libraries were sequenced on the Illumina NovaSeq6000 instrument on the S4 flowcell using a 29 + 8 + 101 sequencing configuration to generate 500 M reads of data per sample.

### 2.8. Analysis of scRNA Sequencing

Raw scRNA-seq data were processed using the 10x Genomics Cell Ranger 6.0.0 software. This pipeline converted Illumina basecall files to fastq format, aligned sequencing reads to the mm10 transcriptome, and quantified the transcript expression in each cell. Cells that passed Cell Rangers quality control measures were subsequently analyzed in R Statistical Software (v3.6.1 and v4.3.2) using the Seurat v4 package [18,24]. For comparison of the LV and RV in the uninjured heart, the sex identity of cells was defined based on their expression of various X- (*Xist*) and Y-linked (*Ddx3y*, *Eif2s3y*, *Gm29650*, *Kdm5d*, and *Uty*) genes. A cell with non-zero expression of Xist and zero expression of our Y chromosome genes was classified as a female cell, whereas a cell expressing one or more of the five Y chromosome genes along with zero expression of Xist was classified as a male cell. Cells that did not meet either of these criteria were treated as having an ambiguous sex identity and removed from the data.

Dimensionality reduction using principal component analysis (PCA) was performed to explore transcriptional heterogeneity. Using heuristics outlined in vignettes for the Seurat software (https://satijalab.org/seurat/vignettes.html, accessed on 5 May 2020), we identified 30 PCs that explained more variability than expected by chance. These PCs were used as input for Seurat’s graph-based clustering program (clustering resolution = 0.5) as well as for Uniform Manifold Approximation and Projection (UMAP) for two-dimensional visualization of cells and clusters. Clusters that displayed gene expression signatures indicative of a non-fibroblast cell type were removed.

We further removed any CFs believed to be derived from the cardiac valves. Since the valves were surgically removed during dissection, this CF subpopulation could not be compared across samples in an unbiased manner. It has previously been reported that CFs expressing *Wif1* localize to the cardiac valves and their adjacent hinge regions [25]. Therefore, we removed cells with counts for *Wif1* > 0.0 from the dataset. We then normalized cell number by randomly selecting cells from each sample, and repeated the above steps for variable gene detection, dimensionality reduction, and clustering. This number corresponded to the number of cells present in the sample with the fewest cells. Outputs from the data normalized to cell number were used for downstream analyses.

### 2.9. Differential Gene Expression Analysis

To calculate differentially expressed genes, we first identified genes that were expressed in at least 10% of cells within at least one cluster and that had a minimum log_2_fold-change of 0.25. *p*-values were determined using the Wilcoxon Rank Sum test implemented through the Seurat FindAllMarkers program. Genes with a Bonferroni-adjusted *p*-value lower than 1 × 10^5^ were defined as differentially expressed.

### 2.10. RNA Velocity and Latent Time Analyses

RNA velocity was examined using Velocyto (v0.17.17) to count spliced and unspliced reads, and scVelo (v0.2.4) was used to compute velocity [26,27]. scVelo was run using the arguments min_shared_counts = 20 and n_top_genes = 2000 for the filter_and_normalize() function, and analyses were carried out using 30 principal components. Velocity and latent time were then calculated using the dynamical model. Cell #2000, which belongs to our Sca1-L cluster, was chosen as the trajectory root. This decision was made based on previous trajectory analyses that place Sca1-L CFs near the origin of the trajectory as well as a decline in Sca1-L CF abundance following MI, which suggests they may differentiate into injury-induced CFs [5].

### 2.11. EGAD Analysis

Meta-analytic bulk co-expression networks for 11 mammals (human, chimpanzee, crab-eating macaque, mouse, rat, rabbit, cow, sheep, goat, dog, and horse) were downloaded from CoCoCoNet [28]. Briefly, for each species, we downloaded RNA-seq datasets with 10 or more samples from SRAdb, totaling hundreds to thousands of samples per species. Co-expression networks for individual datasets were constructed by calculating the Spearman correlation between all gene pairs, followed by ranking the correlation coefficients, with NAs assigned the median rank. Meta-analytic networks for each species are built by aggregating the rank-standardized networks generated from individual datasets.

We evaluated the connectivity of our gene sets in co-expression networks using the run_neighbor_voting function from the EGAD R package [29]. The neighbor-voting algorithm measures the performance of a network in predicting connectivity within gene sets and reports the results as an area under the receiver operator characteristic curve (AUROC). An AUROC of 0.5 indicates random connectivity of genes, while an AUROC close to 1 indicates that genes in our gene set are strongly connected in the co-expression network. AUROC values above 0.8 are generally considered to reflect very high levels of co-expression.

## 3. Results

### 3.1. LV- and RV-Derived CFs Show Distinct Transcriptomic Signatures

We performed bulk RNA-seq of CFs isolated from the LV and RV of healthy 12–16-week-old male *Col1a1-GFP* mice (Figure 1A) [10]. Analysis with the limma R package identified 422 differentially expressed genes between LV- and RV-derived CFs (Figure 1B,C). Among the top hits by *p*-value are numerous fibrosis-related genes, including *Igfbp3*, *Ccl11*, *Vit*, and *Inmt* elevated in the RV and *Col8a1*, *Ctgf*, *Aspn*, and *Postn* elevated in the LV (Figure 1B). Gene set enrichment analysis (GSEA) showed that 83 gene sets were significantly different between ventricles. LV-enriched gene sets were implicated in protein production and mitosis, while those enriched in the RV were related to olfactory signaling and synaptic processes (Figure 1D). These results indicate that LV-derived CFs may be more primed for myofibroblast activation compared with their RV counterparts.

### 3.2. EGAD Analysis Identifies Strong Co-Expression between Genes Differentially Expressed by LV- and RV-Derived CFs

To determine whether the genes identified above are broadly co-expressed (i.e., across various treatments, conditions, etc.), we conducted an Extending ′Guilt-by-Association′ by Degree (EGAD) analysis. This method can predict how strongly a set of genes is co-expressed in a gene co-expression network and reports the results as an area under the receiver operating characteristic curve (AUROC) [29]. An AUROC of 0.5 indicates random connectivity of genes, while an AUROC of 1 indicates perfect co-expression. We initially performed EGAD analysis with a ventricle-specific approach, testing the top 25 upregulated genes in the LV and RV separately. While clusters of co-expressed genes appeared, the full sets of 25 genes did not display strong co-expression. We then selected the clusters of co-expressed genes from both the top LV and top RV sets and combined them to make a third set of 21 genes, representing the top upregulated hits from both ventricles. EGAD analysis of these genes returned an AUROC of 0.971, indicating very strong co-expression of our gene set (Figure 1E). We next studied whether this co-expression was conserved across species using transcriptomic data from various mammals, including humans, chimpanzees, pigs, and crab-eating macaques. The analysis showed strong co-expression across all species tested, including humans with an AUROC of 0.885 (Figure 1F). These findings suggest that the genes differentially expressed by LV- and RV-derived CFs share common regulatory elements, such as transcription factors or epigenetic signatures, and that these ventricle-specific differences are conserved throughout mammalian evolution.

### 3.3. Single-Cell RNA-Seq Reveals Ventricle-Specific Differences between CF Subpopulations in the Uninjured Heart

Given the transcriptomic differences seen through bulk RNA-seq, we sought to explore whether compositional changes in CF subpopulations were driving these differences. To acquire the high-resolution data necessary to answer this question, we performed scRNA-seq on CFs isolated from the LV and RV of uninjured male and female mice (*n* = 4 per ventricle). Transcriptional profiles for 13,755 cells were captured after applying quality control filters (LV, 6548; RV, 7207). Contaminating cell types were then removed, and remaining cells were given female or male identity based on the expression of various X- (*Xist*) and Y-linked (*Ddx3y*, *Eif2s3y*, *Gm29650*, *Kdm5d*, and *Uty*) genes, respectively. The data were downsampled to ensure that all groups (LV female, LV male, RV female, and RV male) contained the same number of cells, resulting in 2032 cells per group and 8128 cells total. CFs with distinct transcriptional states were identified by unbiased clustering using the Seurat R package and visualized using Uniform Manifold Approximation and Projection (UMAP) dimensionality reduction plots.

Our analysis identified eight CF subpopulations within the uninjured ventricles (Figure 2A), many of which reflect clusters first identified by Farbehi et al. [5]. The two largest clusters represent male- and female-dominant subpopulations of *Sca1*-low cells (*Sca1*-LM and *Sca1*-LF, respectively), defined by low *Sca1* expression (Figure 2C,D). CFs with intermediate (*Sca1*-T) and high (*Sca1*-H) *Sca1* expression appeared as smaller clusters, capturing the spectrum of *Sca1*-expressing CFs that has been consistently reported in the literature [5,6,7]. *Postn*-expressing (*Postn*+) and interferon-stimulated (IFNS) CFs were also present in our data, corresponding to the F-Act and IFNS subpopulations reported by Farbehi et al. Finally, we identified three CF subpopulations that were previously undescribed, marked by the expression of *Igfbp3* (*Igfbp3*+), *Mfap4* (*Mfap4*+), and *Gdf15* (*Gdf15*+). Our analysis of the data published by Farbehi et al. shows that CFs with transcriptional signatures similar to these subpopulations exist as subsets of the F-Act and F-Trans clusters. The top differentially expressed genes for these clusters show mixed pro- and anti-fibrotic roles, and GSEA comparisons with other clusters did not reveal any patterns with regard to their function.

While all eight subpopulations were identified in the LV and RV, *Postn*+ and *Igfbp3*+ CFs display ventricle-specific differences. *Postn*+ CFs are more abundant in the LV (478 cells vs. RV: 280 cells), whereas *Igfbp3*+ CFs are more abundant in the RV (415 cells vs. LV: 234 cells) (Figure 2B). Comparisons with Farbehi et al. and other published datasets suggest that *Postn+* CFs are pro-fibrotic and primed for differentiation into injury-induced CFs. In contrast, there is little information regarding *Igfbp3*+ CFs and their role in health and disease, though a literature review points to a mixed pro- and anti-fibrotic phenotype. These findings are consistent with our bulk RNA-seq data that imply a stronger fibrotic response in the LV following cardiac injury.

### 3.4. Single-Cell RNA-Seq Reveals Ventricle-Specific Differences between CF Subpopulations in the Pressure-Overloaded Heart

The baseline differences in LV and RV CF subpopulations, and their relevance to the fibrotic response, prompted further investigation into ventricle-specific CF heterogeneity following injury. We compared LV- and RV-derived CFs under pressure overload conditions using transverse aortic constriction (TAC) and pulmonary artery banding (PAB) as injury models for the LV and RV, respectively. After two weeks of stimulus, CFs were harvested and used for scRNA-seq (n = 3 per sample), with male and female samples prepared and sequenced separately. Echocardiography was performed to assess cardiac function and ensure that mice with TAC and PAB had comparable extents of injury (Appendix A). After quality control, aggregated data captured the transcriptional profiles of 43,938 cells. Removal of contaminating cell types and subsequent downsampling resulted in 2582 cells per group and 20,656 cells in total.

Unbiased clustering using the *Seurat* R package identified 12 distinct clusters (Figure 3A and Appendix A). All CF subpopulations in our baseline data also emerged here, though male and female *Sca1*-L CFs combined into one cluster. The greater resolution of this experiment generated a distinct cluster for CFs expressing immediate early genes (IEGs). It also separated CFs with intermediate *Sca1* expression into three groups: *Sca1*-T1, *Sca1*-T2, and *Ccl2*+. The *Ccl2*+ subpopulation has a particularly unique gene signature, marked by strong and specific expression of immune-relevant genes such as *Ccl2*, *Ccl7*, and *Gdf15*. A group of CFs marked by *Inmt* (*Inmt*+) became distinguishable from the *Mfap4*+ cluster, for which *Inmt* is also a top marker. Finally, we report a cluster of CFs marked by expression of *Thbs4*, *Comp*, and *Fmod* (*Thbs4*+) that is more abundant in our pressure overload samples (Figure 3B).

Picosirus red staining suggests a comparable degree of injury between our TAC and PAB mice (Figure 3D). Interestingly, comparisons of LV- and RV-derived CFs reveal diverging injury responses for the two ventricles. The LV shows greater differentiation into known pro-fibrotic clusters, developing a larger subpopulation of *Postn*+ and *Thbs4*+ CFs (Figure 3C,E). In contrast, the injured RV shows an expansion of *Igfbp3*+ and *Inmt*+ CFs, which is absent in the LV (Figure 4A,B).

EGAD analysis of the top 25 marker genes from each of our CF subpopulations found that most clusters had AUROC scores greater than 0.9 (Appendix A). The IFNS cluster scored particularly high, with an AUROC of 0.984. Overall, such strong co-expression of the marker genes defining CF clusters supports the conclusion that they are transcriptionally distinct states that provide meaningful info about CF biology.

### 3.5. RNA Velocity Analysis Identifies a Trajectory toward Thbs4+ CFs

To explore potential trajectories between CF subpopulations, we next performed RNA velocity analysis. We first reanalyzed the data using Velocyto to acquire RNA counts for spliced and unspliced transcripts [26]. Unbiased clustering using spliced RNA counts for 20,360 cells (2545 cells per group) identified 11 distinct clusters (Figure 5A). These clusters were similar to those identified in the original analysis, with the exception of *Thbs4*+ CFs, which became a subset of the *Postn*+ cluster. We then performed latent time analysis using the *scVelo* python package [27]. Cells in our *Sca1*-L cluster were chosen as the trajectory root based on previous trajectory analyses that place *Sca1*-L CFs near the origin as well as a greater than expected decline in *Sca1*-L CF abundance following MI, which suggests they may be differentiating into injury-induced CFs. While the gene signature of *Sca1*-H CFs suggests a progenitor phenotype, trajectory analysis using multiple methods consistently places them at later pseudotime positions, suggesting they represent a more differentiated CF state. Latent time values overlaid on the UMAP show *Thbs4*+ CFs as the most differentiated subpopulation, with *Postn*+ CFs close behind and feeding into the *Thbs4*+ cells (Figure 5B). Considering the enrichment of these cells in the LV, our findings support the conclusion that LV-derived CFs are further along the differentiation trajectory toward *Thbs4*+ CFs. In contrast, *Igfbp3*+ and *Inmt*+ CFs, which expand in the injured RV, were not identified as highly differentiated subpopulations.

### 3.6. Males and Females Display Differences in CF Subpopulation Size and Injury Dynamics

Separating the data by male and female identity revealed sex differences in the abundance of several clusters (Figure 6A). Both the uninjured and pressure overload datasets show that healthy females (uninjured and sham groups) have a larger subpopulation of *Postn*+ CFs compared with males. However, the abundance of both *Postn*+ and *Thbs4*+ CFs is greater in males following injury (Figure 6B,C), suggesting that males start in a less fibrotic state but mount a more robust injury response. Differences in *Igfbp3*+ CFs are also apparent (this subpopulation is larger in female RV samples from sham and injury conditions compared with males). This cluster showed no differences in male and female LV samples (Appendix A), consistent with the idea that *Igfbp3*+ CFs are an RV-dominant subpopulation. Finally, we report that the Sca1-L subpopulation appears to be substantially larger in male sham mice compared with all other groups.

## 4. Discussion

Here, we present bulk and scRNA-seq data on >28,000 CFs from the LV and RV of uninjured and pressure-overloaded murine hearts. We identified 422 differentially expressed genes between LV- and RV-derived CFs, with GSEA highlighting gene sets for protein production and mitosis as more highly expressed in the LV of healthy (non-injured) hearts. The top hits from this gene set exhibit strong transcriptomic co-expression that is conserved across mammals, including humans, indicating that these differences may be relevant in clinical practice.

Further analysis of the various CF subgroups from healthy cardiac tissue using scRNA-seq identified ventricle-specific increases in the abundance of *Postn*+ and *Igfbp3*+ clusters in the LV and RV, respectively. The postulated role of *Postn*+ CFs as a pro-fibrotic subpopulation that is primed for differentiation, together with the GSEA results, suggests the healthy LV is primed to be a more pro-fibrotic environment than the healthy RV, with greater numbers of CFs located further along a myofibroblast differentiation trajectory. Indeed, scRNA-seq of LV- and RV-derived CFs following pressure overload injury revealed that the *Postn*+ and injury-induced *Thbs4*+ clusters grew substantially more in the LV than the RV. In contrast, *Igfbp3*+ and *Inmt*+ CFs selectively expanded in the pressure-overloaded RV. These findings demonstrate that the LV also mounts a stronger fibrotic response to pressure overload injury than the RV. RNA velocity analysis further identified a trajectory wherein *Postn*+ CFs differentiate into the *Thbs4*+ subpopulation, confirming that LV-derived CFs are shifted further along the *Thbs4* differentiation trajectory in the uninjured heart and validating the role of *Postn*+ CFs as precursors for an injury-induced subpopulation.

These data highlight the need for a better understanding of *Igfbp3*+ and *Inmt*+ CF subpopulations in the heart and their potential role(s) in cardiac injury and repair. Our analysis of the scRNA-seq dataset published by Farbehi et al. on cardiac-derived *Pdgfra*+ cells shows that both subpopulations are present as subsets of what they referred to as the F-Trans cluster, a subpopulation that is predicted to be partway along the differentiation trajectory of CFs to myofibroblasts [5]. The top markers for our *Igfbp3*+ CFs include *Apoe*, *Fgl2,* and *Igfbp3*, which shows an 11.85-fold increase in expression compared with other clusters. The IGFBP3 protein is the primary carrier of IGF-1 in the serum, thus allowing it to potently regulate systemic and local IGF-1 activity [30]. However, this protein also exhibits IGF-independent effects through various cell-surface receptors, including TMEM219 and TGF-β receptors I, II, and V [31,32]. Transcriptomic data show that TFG-β-receptor V, in particular, is predominantly expressed by CFs and broadly expressed across all CF subpopulations [33]. Since TGF-β signaling is a well-known activator of CF differentiation and proliferation [34], these findings suggest that IGFBP3 may contribute to the initiation and progression of the fibrotic response. This idea is supported by a recent paper studying cultured neonatal CFs transfected with *Igfbp3*-siRNA after 24 h of TGF-β1 stimulation. CFs with *Igfbp3* knock-down showed reduced migration and proliferation, as well as lower expression of αSMA and Collagen I, suggesting a role in activating CFs during injury [35].

In contrast, *Inmt*+ CFs, marked by *Inmt*, *Sfrp2*, *Ptn*, and *Vegfd*, may play a role in downregulating the fibrotic response. *Inmt* shows the greatest upregulation, with a 3.5-fold increase over other CFs. This gene has been reported as downregulated in atrial fibrillation and MI conditions [36] as well as in infarcted compared with remote tissue [11]. Transcriptomic data reveal that *Inmt* expression in the heart is primarily restricted to CFs, suggesting its role in the maintenance and regulation of the cardiac ECM [33]. INMT is an intracellular methyltransferase most noted for its involvement in tryptamine N-methylation and xenobiotic clearance [37,38]. However, it also demonstrates an anti-proliferative role in many cell types, including skin- and lung-derived fibroblasts, with higher *Inmt* expression linked to cellular senescence and apoptosis and downregulation implicated in cancer [37,38,39]. This relationship supports our findings that the RV fibroblast landscape is relatively less activated compared with that of the LV, particularly following injury. It may further explain why INMT expression is greater in remote regions following MI [11], where little scarring, if any, is necessary. A previous study discovered that *Inmt* overexpression in prostate cancer cells inhibited effectors of Wnt and TGF-β pathways, including SMAD4 [37]. Given the central importance of TGF-β signaling to myofibroblast activation, this finding may predict attenuated fibrotic responses in the injured RV. Paradoxically, *Inmt* overexpression also stimulated the p38 MAPK pathway, which promotes myofibroblast activation and ECM remodeling [37,40]. While much work remains to be done to characterize the role of *Igfbp3*+ and *Inmt*+ CFs in the heart, these mechanistic insights provide initial hypotheses for exploring their contributions to the fibrotic response.

The *Wif1*-expressing CFs reported by several other groups were also identified in both our uninjured and pressure overload scRNA-seq datasets. However, immunofluorescence and RNAscope data from Muhl et al. identify *Wif1*+ CFs as localized to the cardiac valves and adjacent hinge regions [24]. While our isolation protocol involves removal of the cardiac valves, we did not control for dissection of the hinge regions. Accordingly, we chose to remove cells that belonged to this cluster from our analysis as their abundance in each sample may be driven by tissue dissection rather than ventricle identity or injury status.

Separating the data by sex revealed additional heterogeneity in injury-induced subpopulations. While *Postn*+ and *Thbs4*+ CFs are more abundant in females in the uninjured heart, these subpopulations display greater expansion in males following pressure overload. As such, males appear to develop a more substantial fibrotic response to pressure overload injury, providing a potential physiologic mechanism for the increased cardiac fibrosis observed in males from murine and human studies [41]. These results contradict those published by McLellan et al., who report a larger *Thbs4*+ cluster in females after Angiotensin II (ANG II) infusion injury [7]. The disconnect between our results may be due to the use of a different injury model; while both models cause increased afterload, Angiotensin II has additional pharmacologic effects on the heart that contribute to CF activation and production of Tgfb1 [42,43,44,45,46]. Furthermore, there are sex differences in the physiological response to Angiotensin II, particularly regarding the development of hypertension. Males experience a greater increase in blood pressure compared with females, which appears to be driven by differences in sex hormones [47]. Notwithstanding the differences seen in *Thbs4*+ CF subgroups, McLellan et al. report that females have less fibrosis in the LV, which is consistent with our results and the overall body of literature on sex-dependent differences in fibrosis.

We also note that the *Sca1*-L subpopulation is more abundant in male shams compared with all other groups. This may be due to the low abundance of *Postn*+ CFs present in this group, resulting in the other dominant CF subpopulations, including *Sca1*-L, making up a greater proportion of the total CF population.

## 5. Conclusions

In summary, our study demonstrates that LV- and RV-derived CFs display subpopulation differences in the healthy heart that may cause their diverging responses to pressure overload. These findings may also help guide our understanding and treatment of complex cardiac conditions such as Arrythmogenic Right Ventricular Cardiomyopathy, where differences in the extent and type of fibrotic deposition (interstitial versus replacement) and right versus bilateral ventricular involvement between patients are poorly understood. In addition, there are notable transcriptomic differences across biological sex, suggesting a sexually dimorphic response to cardiac injury. The most effective interventions for cardiac fibrosis will have to accommodate these differences, with treatments tailored to the at-risk ventricle and sex of the patient. Continued study with the latest scRNA-seq tools will enable analysis of the geographic and spatiotemporal characteristics of CF subpopulations for a more complete understanding of their functions at baseline and in the injured heart.

## Figures and Tables

**Figure 1 cells-13-00327-f001:**
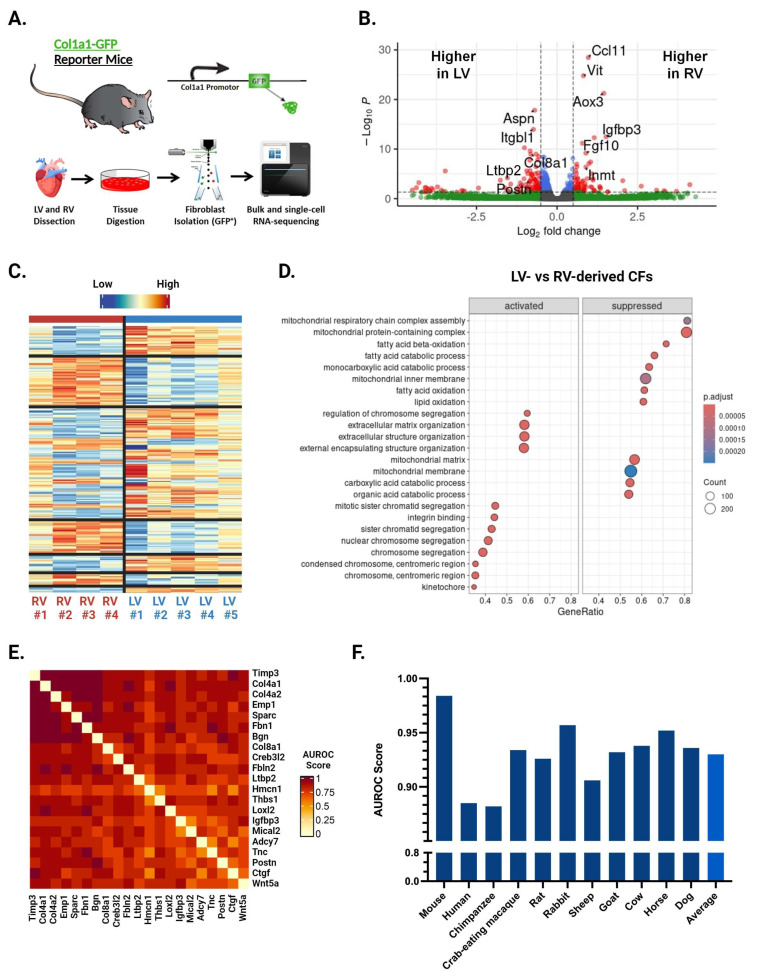
Bulk RNA sequencing and EGAD results from LV- and RV-derived CFs. (**A**) Schematic of methods used to isolate cardiac fibroblasts (CFs) for bulk and single-cell RNA sequencing (scRNA-seq). (**B**) Volcano plot of genes exhibiting differential expression between LV- and RV-derived CFs. Top fibrosis-related genes by *p*-value are labeled. Color represents whether a gene surpasses the cut-off for p-value (*p* < 0.05), fold change (|log2FC| > 0.5), both, or neither. (**C**) Heatmap of genes exhibiting differential expression between LV- and RV-derived CFs. (**D**) Top differentially expressed gene sets identified by GSEA. Labels for “activated” and “suppressed” are with respect to the LV. (**E**) Heatmap showing co-expression scores for the top 21 differentially expressed genes between LV- and RV-derived CFs. Darker and lighter colors represent higher and lower co-expression scores, respectively. Values for genes being compared against themselves have been set to 0. (**F**) Area under the receiver operator characteristic curve (AUROC) values across 11 different species for the top 25 genes differentially expressed by LV- and RV-derived CFs.

**Figure 2 cells-13-00327-f002:**
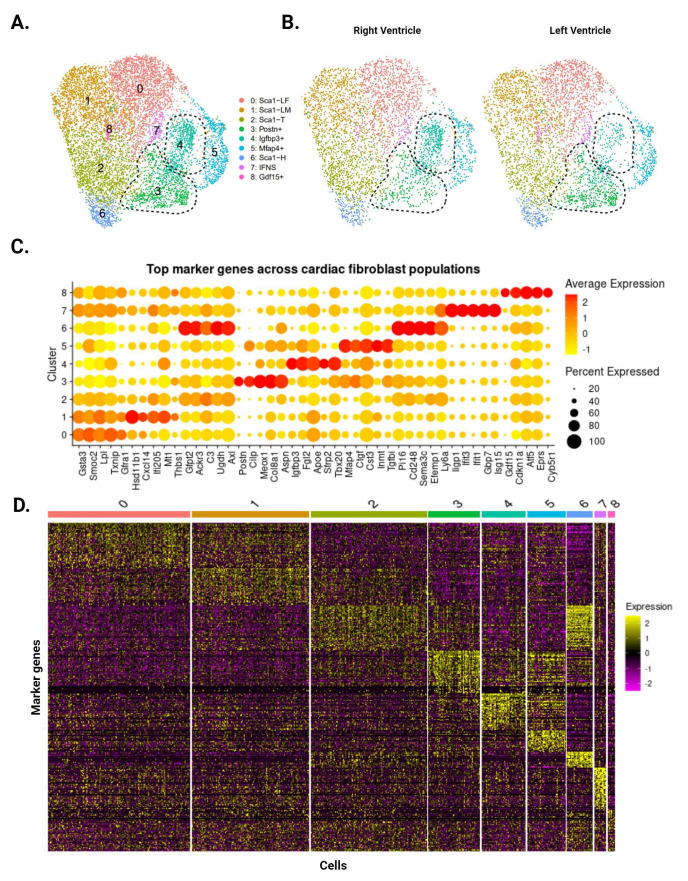
Single-cell RNA sequencing of CFs from uninjured LV and RV tissue. (**A**) Aggregate UMAP of all CFs identified in LV and RV samples. Dotted lines demarcate the *Postn*+ and *Igfbp3*+ subpopulations. (**B**) CFs separated according to ventricle of origin. *Postn*+ CFs are more prominent in the LV, while *Igfbp3*+ CFs are more prominent in the RV. (**C**) Dotplot of top five upregulated genes for each subpopulation. Color indicates expression level and dot size represents the percentage of cells expressing each gene. (**D**) Heatmap of the top 50 marker genes for each subpopulation. Individual cells are ordered along the x-axis, while marker genes are ordered along the y-axis. Color represents the expression level of each gene.

**Figure 3 cells-13-00327-f003:**
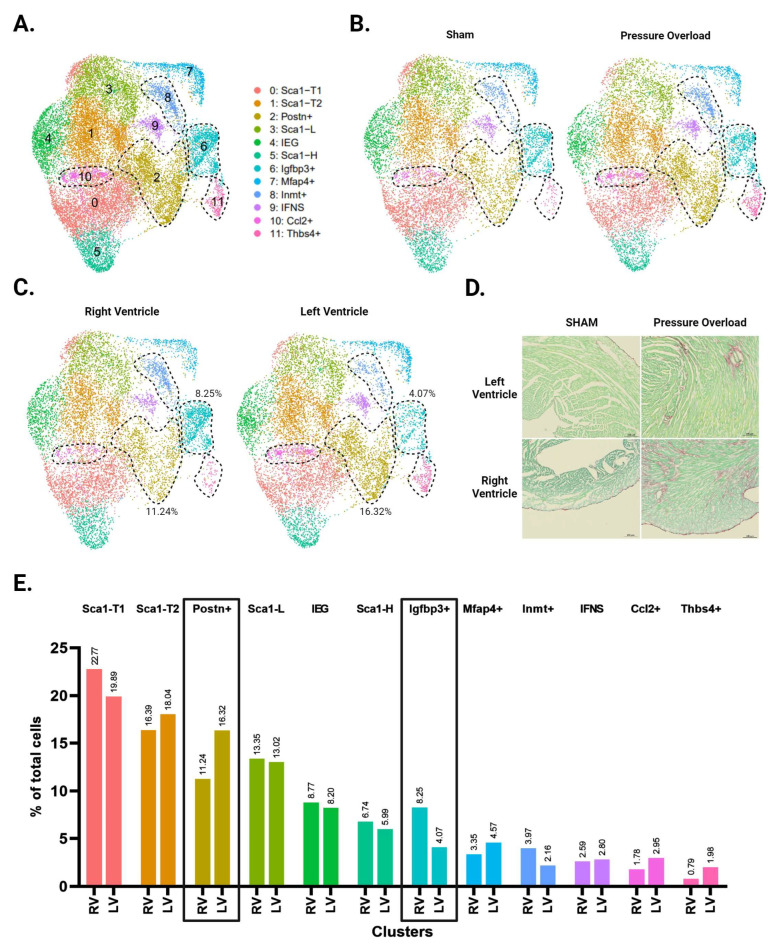
Single-cell RNA sequencing of CFs from LV and RV tissue following pressure overload injury. (**A**) Aggregate UMAP of all CFs in LV and RV samples from both male and female pressure overload experiments. (**B**) CFs separated by injury status. (**C**) CFs separated according to ventricle of origin. (**D**) Representative fibrosis staining of whole adult hearts with transverse aortic constriction (TAC), pulmonary artery banding (PAB) injury, and their respective sham surgeries. (**E**) Proportion of cells in each subpopulation split by ventricle. Subpopulations that show increased abundance in the LV or RV are highlighted by black boxes. (**A**–**C**) CF subpopulations displaying ventricle-specific differences in abundance are demarcated by dotted lines.

**Figure 4 cells-13-00327-f004:**
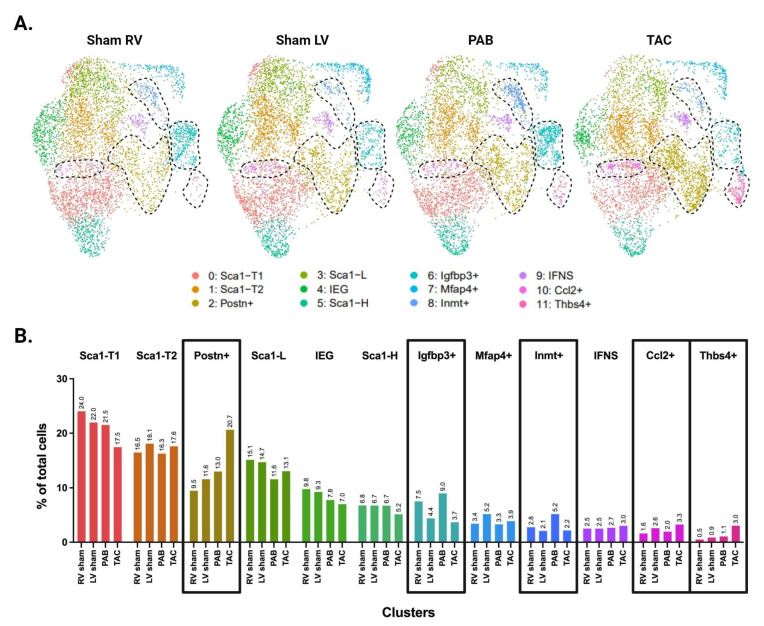
Distribution of CF subpopulations across ventricle and injury status. (**A**) UMAP of CFs separated by ventricle and injury status. Injury-related CF subpopulations are outlined. (**B**) Proportion of cells in each subpopulation split by ventricle and injury status. (**A**,**B**) Subpopulations that show increased abundance in the LV or RV are demarcated by dotted lines in panel **A**, or black boxes in panel **B**.

**Figure 5 cells-13-00327-f005:**
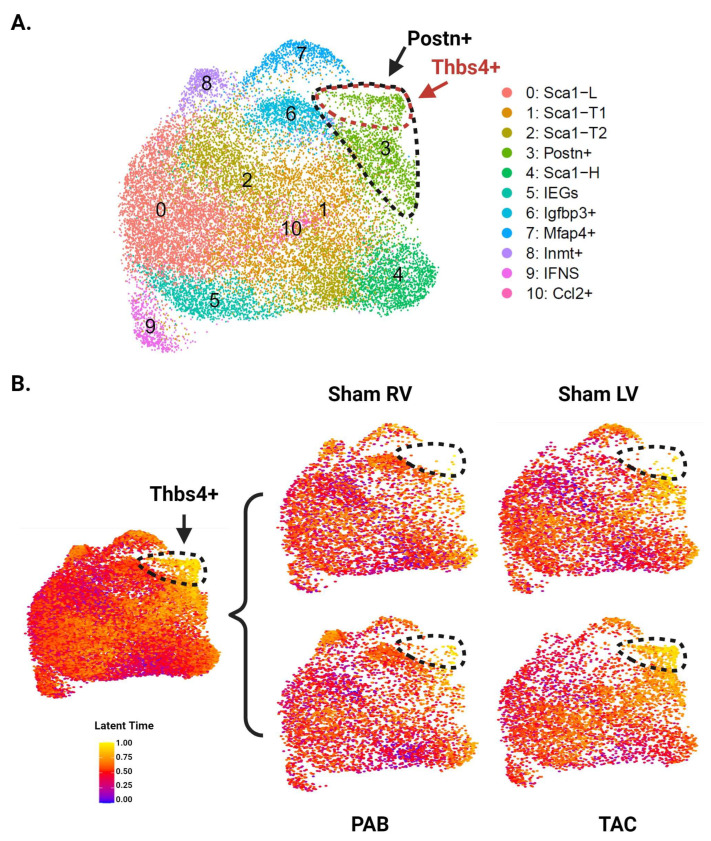
RNA velocity analysis of CFs from LV and RV tissue following pressure overload injury. (**A**) Aggregate UMAP of all CFs from pressure overload experiments after reanalysis with Velocyto outputs. *Postn*+ CFs are demarcated by the black dotted line, and *Thbs4*+ CFs are demarcated by the red dotted line. (**B**) UMAP colored by latent time values of each cell and split by ventricle and injury status. The color of each cell indicates its position along the predicted differentiation trajectory, with yellow and blue representing the latest and earliest positions, respectively. CFs at the latest positions along this trajectory—corresponding to the *Thbs4*+ subpopulation—are demarcated by black dotted lines.

**Figure 6 cells-13-00327-f006:**
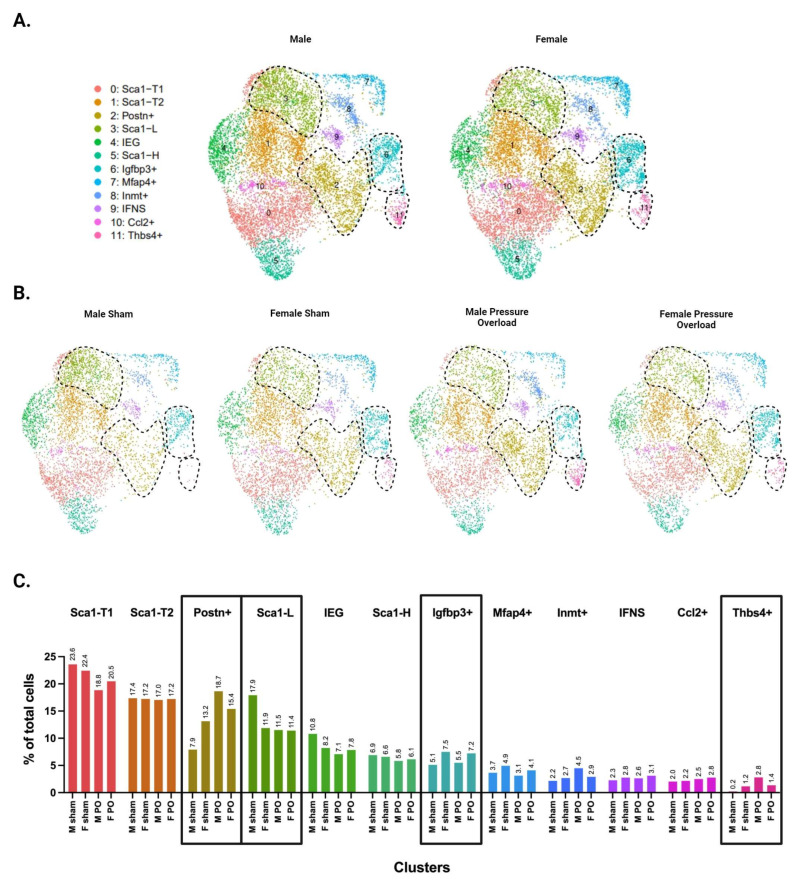
Single-cell RNA sequencing of male versus female CFs following pressure overload injury. (**A**) UMAPs of CFs separated by biological sex. (**B**) UMAPs of CFs separated by biological sex and injury status. (**C**) Proportion of cells in each subpopulation split by biological sex and injury status. (**A**–**C**) CF subpopulations displaying sex-specific differences in abundance are demarcated by black dotted lines in panels **A** & **B** or by black boxes in panel **C**.

## Data Availability

The sequencing data presented in this study have been deposited in the Gene Expression Omnibus (GEO) database, and are available under accession codes GSE255542 (bulk RNA-seq) and GSE254914 (single-cell RNA-seq).

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
