# Peer review of "Defining Transcriptomic Heterogeneity between Left and Right Ventricle-Derived Cardiac Fibroblasts"

_cells, 2024, doi:10.3390/cells13040327_

Round 1

Reviewer 1 Report

Comments and Suggestions for Authors

The author collected cardiac fibroblast of RV and LV separately from healthy and pressure overload mice hearts for bulk RNA-Seq and single cell RNA-Seq, identified RV and LV specific genes and specific CF population (Postn+ vs Igfbp 3+) in each ventricle. Comparing the RV and LV CFs before and after the PBC and TAC models showed different responses of LV and RV CFs to the injuries. Overall the data is sufficient in most facets of the authors' claim, however some histology, immunostaining and elucidating data representation can be included to improve the manuscript

Here are some questions for this manuscript:

Major

  1. Would the author show the FACS plot for sorting of fibroblast from Col1a1-GFP mice? Can this method collect all the fibroblast in the ventricle or the major population?  Would be better to show some co-staining of GFP and fibroblast maker in the ventricle

  2. Fig 1E, why the correlation between the same gene is the lowest, i.e. why the color on the diagonal is the brightest? 

  3. It would be better to show the percentage of Postn+ or Igfbp3+ in RV and LV to indicate their ventricle specific difference, and may the author show some verification of the CFs in LV and RV respectively by staining?

  4. Could the author also separate the PBC and TAC CFs percentage data into RV and LV in Figure 4B.

  5. Could the author provide some histological or immunostaining data to show the fibrosis after the PBC and TAC model?

Minor issue

  1. FIg 1E, Genes would be better categorized into subsets that are highly expressed in RV and LV respectively.

  2. Figure 3D missing

  3. Please include scale bars for Figure 1E and Fig S4.

  4. Please include the cluster identities in Figure 4.

  5. Fig 4B, all boxes are black, not as the legend described.

  6. Line 391, I think the author mean Thbs4 rather than Thbs1

  7.  Line 400, Supplementary Figure 4

Reviewer 2 Report

Comments and Suggestions for Authors

In the present manuscript by Dewar and colleagues, murine fibroblasts from the left and right ventricle were compared on gene expression level. The aim of this study is highly interesting and the experimental approach is scientifically sound. The results are clearly presented and discussed appropriately. Please find below my minor comments:

In the abstract and introduction, it becomes not directly clear to the reader, that the authors performed experiments with mice/ murine material and not with human material.

The mouse strain should be described in the methods section (e.g. in Section 2.1)

Figure 1D: the font size is very small

Figure 2D should be explained in more detail and the y axis should be labelled with the gene names

Figure 4B: The authors should either correct the colour of the boxes or correct the figure caption (ll.397-398)

Reviewer 3 Report

Comments and Suggestions for Authors

The work by Dewar et al focuses on cardiac fibrosis and potential influences of ventricle specificity, gender and disease state in the population makeup of those chambers. Their work adds to a growing body knowledge highlighting the heterogeneity of cardiac fibroblast populations and consequently their potential differential response to disease states. The manuscript is well conceived and the results are appropriately presented and discussed.

While recognizing that the topic is tremendously complex, and current knowledge is only sufficient to begin a narrative, the discussion could perhaps be expanded slightly further to include more perspective on potential functional implications of their findings and how they may relate to disease management. If the authors should chose to do so, perhaps contextualization of their results with knowledge from diseases that seem to have ventricle-specific manifestations, such as Arrhythmogenic Right Ventricular Cardiomyopathy (ARVC), where the right ventricle appears to have a large contribution to the disease state, could be interesting.

Nonetheless the above should not preclude the acceptance of the manuscript in its current form.

Round 2

Reviewer 1 Report

Comments and Suggestions for Authors

After revision, the manuscript has improved and can be published, the topic is interesting and informative for future researchers.